# SLayR: Scene Layout Generation with Rectified Flow

## Abstract

We introduce SLayR, **S**cene **Lay**out Generation with **R**ectified flow, a novel transformer-based model for text-to-layout generation, which can integrate into a complete text-to-image pipeline. SLayR addresses a domain in which current text-to-image pipelines struggle: generating scene layouts that are of significant variety and plausibility, when the given prompt is ambiguous and does not provide constraints on the scene. In this setting, SLayR surpasses existing baselines, including LLMs. To accurately evaluate the layout generation, we introduce a new benchmark suite, including numerical metrics and a carefully designed repeatable human-evaluation procedure that assesses the plausibility and variety of images that are generated. We show that our method sets a new state of the art for achieving high plausibility and variety simultaneously, while being at least $3\times$ times smaller in the number of parameters.

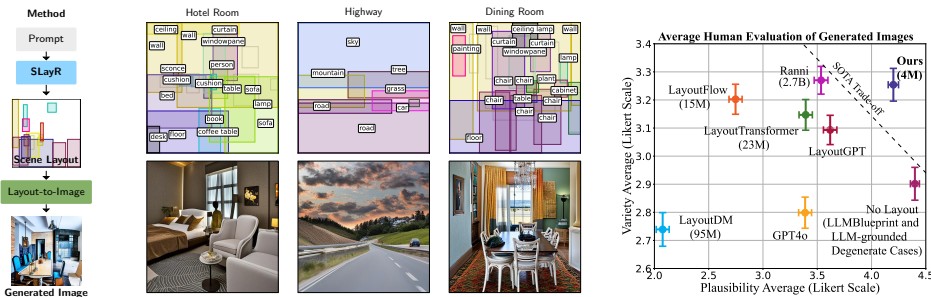

Figure 1: **Left:** We introduce **SLayR**, a method for scene layout generation via rectified flow. **Middle:** SLayR generates scene layouts for unconstrained prompts, which can be rendered using a layout-to-image generator. **Right:** Our method sets a new state of the art in generating more varied and yet plausible scenes than baselines, including LLMs.

## 1 Introduction

Recent advances in text-to-image modeling have focused on training denoising diffusion models (Sohl-Dickstein et al., 2015; Ho et al., 2020; Song et al., 2022) to generate images from a prompt encoding and image noise (Ramesh et al., 2021; Rombach et al., 2022; Saharia et al., 2022; Esser et al., 2024; Zhang et al., 2023a; Sauer et al., 2024), as well as incorporating finer-grained control modalities (Hudson et al., 2023; Kwon et al., 2023; Park et al., 2023; Zhang et al., 2023c; Luo et al., 2024; Shen et al., 2024; Mishra & Subramanyam, 2024; Wu et al., 2023b). Building upon these advancements, prior works have demonstrated the editability and interpretability advantages of a multistage text-to-layout-to-image model, where the user can view and manipulate an intermediate layout consisting of bounding boxes for object-level scene elements (Lian et al., 2024; Feng et al., 2023; Zhou et al., 2024; Gao et al., 2024; Yuan et al., 2024; Öcal et al., 2024; Aguina-Kang et al., 2024). These works use LLMs as text-to-layout generators, and focus on parsing multi-object prompts (e.g. "two dogs next to a cat"). However, a closer inspection reveals that these models do not generate high variety (see Fig. 1, right) or collapse entirely (see Fig. 2), when presented with prompts that have few constraints and leave a high degree of ambiguity. We see this as a critical problem: the models in these cases fail to present knowledge about the structure of scenes as they cannot rely on the

prompt for specific information.

This motivates us to propose SLayR, a novel lightweight text-to-layout generation model for expanding unconstrained prompts (e.g. "a park", "a beach") into a variety of plausible scene layouts (see Fig. 1, left and middle). Given a CLIP (Radford et al., 2021) embedding of a global scene prompt, we generate the layout using rectified flow (Liu et al., 2022), with a Diffusion Transformer (DiT) (Peebles & Xie, 2023). As unconstrained text-to-layout generation for general images has not been explored before, we assess our layout's plausibility and variety against both LLM-centric baselines and adapted UI/document generation. The experiments show that our method produces a very high variety, while achieving state-of-the-art plausibility in spatial arrangement.

Next, we combine our generated layouts with available layout-to-image generation models (Wang et al., 2024; Li et al., 2023; Xie et al., 2023; Lian et al., 2024) to create a complete text-to-image pipeline. We show that the generated images achieve the highest scores in CMMD (Jayasumana et al., 2024), FID (Heusel et al., 2018), KID (Bińkowski et al., 2021), and HPSv2 (Wu et al., 2023a) compared to the baselines. As true assessment of the image content is only possible by humans, we introduce a comprehensive and repeatable human-evaluation study. The ratings show that our model yields the state-of-the-art trade-off in generating images that are both diverse and plausible. In addition, our pipeline is significantly more lightweight than baselines and can be conditioned on partial layouts and directional constraints, while also providing the ability to edit layouts.

In summary, our contributions are: **1)** we introduce the first model for rectified flow-based text-to-layout generation and show that it produces a large variety of highly plausible layouts for challenging unconstrained prompts, **2)** we establish a well-designed human-evaluation study that can be repeated by others, and **3)** demonstrate that integrating our method into a complete text-to-layout-to-image pipeline yields state-of-the-art in achieving variety and plausibility together. See our supplement to access source code.

## 2 Related Work

**LLMs in Scene Layout Generation.** Prior works in 2D layout generation leverage LLMs to parse multi-object prompts into layouts, typically leveraging in-context learning (Lian et al., 2024; Gani et al., 2024; Feng et al., 2023; 2024). Querying these models with unconstrained prompts frequently yields degenerate solutions without meaningful layout information (See Fig. 2). Given that LLM-grounded Diffusion (Lian et al., 2024) and LLM Blueprint (Gani et al., 2024) degenerate in 90% or more cases, we do not evaluate them further. Results on LayoutGPT (Feng et al., 2023) and Ranni (Feng et al., 2024) are provided. To control for the shift to the unconstrained prompt domain, we also adapt the prompt template from (Lian et al., 2024) with in-context examples from our domain, and encouragement of chain-of-thought reasoning (Wei et al., 2023), to meaningfully assess an LLM's capabilities for this task. For the underlying LLM, we use GPT4o (OpenAI et al., 2024)

**Adapting UI Generation.** Our task of scene layout generation is distinct from User Interface (UI) generation: scene and object captions are from open sets, whereas UI layouts lack global captions and have labels from a small fixed set. Nevertheless, they can serve as interesting baselines, and we adapt several of these models using their conditional generation capabilities. We use LayoutTransformer (Gupta et al., 2021) as a representative for autoregressive transformer approaches, which completes a partial sequence of object bounding boxes to form an image layout. LayoutFormer++ (Jiang et al., 2023) extends LayoutTransformer with added conditioning, but this is not the focus of our assessment of adapted UI generation, and thus it is a redundant baseline. We also adapt LayoutDM (Inoue et al., 2023) and LayoutFlow (Guerreiro et al., 2024) as representative baselines for diffusion-based methods for UI generation (Zhang et al., 2023b; Chai et al., 2023; Levi et al., 2023). For GAN-based approaches (Li et al., 2019), while LayoutGAN++ (Kikuchi et al., 2021) supports inter-bounding-box relationships, the Lagrange multiplier constraint formulation cannot be adapted to support global conditioning. In contrast to our method, UI generation models by design do not extend into the open world scenario.

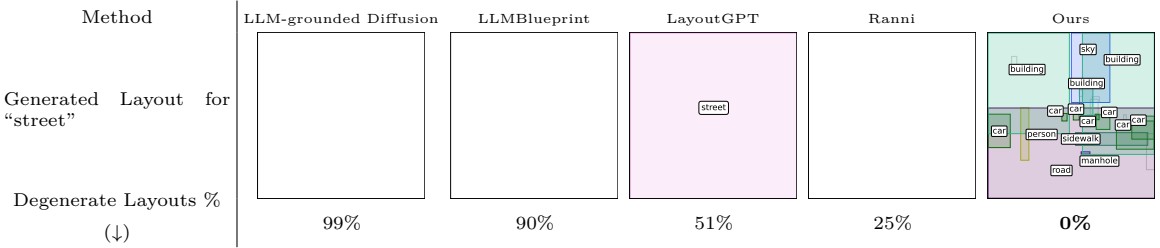

Figure 2: Degenerate layouts (where zero or one trivial bounding box is present) for the prompt "street" from LLM-grounded Diffusion (Lian et al., 2024), LLM Blueprint (Gani et al., 2024), LayoutGPT (Feng et al., 2023), and Ranni (Feng et al., 2024) vs. our layouts. The bottom shows percentages of degenerate layouts from our unconstrained prompt benchmark (See Sec. 4). [We generate layouts in Figure 2, by using the methodology in Sec 4, paragraph Dataset: we query each baseline on 20 scene category labels (e.g., "street", "roof garden") present in ADE20K, collecting 30 sample layouts per category (600 samples total), i.e, multiple samples are generated. We define a layout as degenerate if it (i) contains no bounding boxes, or (ii) contains a single bounding box that occupies the entire frame, with the scene category name as its label. These layouts are degenerate in the sense that they do not provide any meaningful scene structure information to the layout-to-image generator, making text-to-layout generator superfluous to generating the image. Degeneration rates do *not* rely on ADE20K annotations as ground truth. The criterion is based solely on the structure of the generated layout. Under this definition our method achieves a 0% degeneration rate: none of the 600 generated layouts are degenerate (each contains at least 2 bounding boxes).] As visible, LLM approaches for constrained prompts do not generalize to the unconstrained setting. [The high degeneration rates of off-the-shelf LLM methods motivates us to construct a stronger LLM-baseline with in-context examples of ADE20K layouts and chain-of-thought reasoning, referred to as the 'GPT4o' baseline in the text. This reduces degeneration to 0%. However, as shown in Table 1, even with this prompt engineering, LLM-based methods produce layouts that are less aligned with the target distribution than our flow-based approach.]

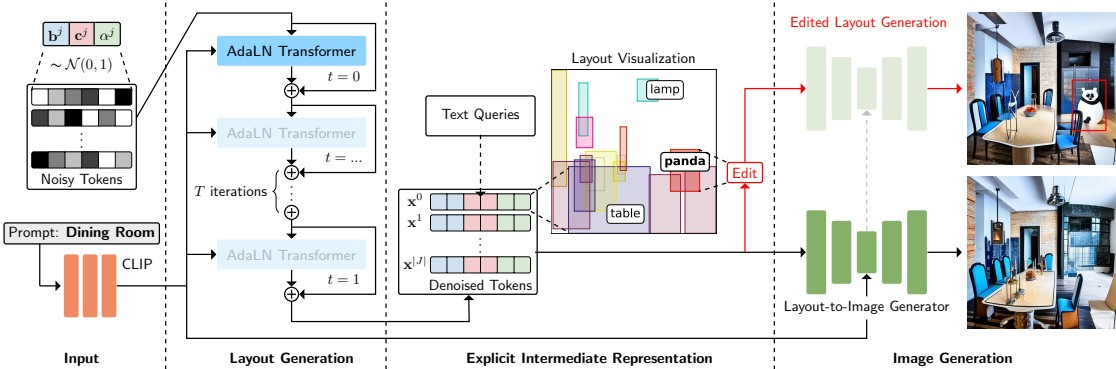

Figure 3: **Method Overview.** Our layout generation model takes a set of noisy tokens and a prompt encoded as a global CLIP embedding as input. The tokens are partitioned into bounding box information $\mathbf{b}^j$, reduced CLIP embeddings $\mathbf{c}^j$, and opacities $\alpha^j$, with $j$ being the object index. The tokens are then subsequently denoised from $t = 0$ to $t = 1$ using a transformer. For visualization purposes, the user can query the generated layout with labels and edit boxes by adding, moving or removing them. Finally, the generated layout is passed through an off-the-shelf layout-to-image generator.

**Rectified Flow.** Diffusion modeling has inspired numerous variants and improvements, one of which is rectified flow (Liu et al., 2022). Prior works on the related text-to-image generation task (Liu et al., 2024; Esser et al., 2024). An initial ablation on DDIM (Song et al., 2022), indicates that rectified flow outperforms traditional diffusion approaches (Ho et al., 2020) in this setting. See the supplement for details.

**Layout-to-Image Generation**. We demonstrate that SLayR integrates well into downstream conditional diffusion models to form a complete text-to-image pipeline, with the added benefits of an interpretable

and controllable intermediate layout phase. To control for the effect which the image generator has on the final generated image, we evaluate our layouts across multiple layout-to-image models. Although there are a wide variety of such models, (Chen et al., 2023; Yang et al., 2022; Zhao et al., 2019; Sylvain et al., 2020; Bar-Tal et al., 2023; Xiong et al., 2024) we select four which are publicly available and have been used successfully with LLM-driven layouts (Lian et al., 2024; Feng et al., 2023) or have shown SOTA performance: InstanceDiffusion (Wang et al., 2024), GLIGEN (Li et al., 2023), BoxDiff (Xie et al., 2023), and LMD+ (Lian et al., 2024).

## 3 Method

The central part of our work is the text-to-layout generation module, which we combine with the existing layout-to-image generators to form a complete text-to-image pipeline. An overview is provided in Fig.3 3, and we explain the details below.

**Layout Representation.** We start with defining a scene representation as the basis for our generative architecture. A training sample $(\mathbf{x}, P)$ is composed of a global image caption prompt $P$ and a set of $J$ object tokens $\mathbf{x} = \{\mathbf{x}^j \in \mathbb{R}^{d+5}\}_{j \in J}$ . The token representation of any single object is composed of

$$\mathbf{x}^j = (\mathbf{b}^j \parallel \mathbf{c}^j \parallel \alpha^j), \tag{1}$$

where $\mathbf{b}^j = (x^j, y^j, w^j, h^j) \in \mathbb{R}^4$ encodes the bounding box coordinates, $\mathbf{c}^j \in \mathbb{R}^d$ is a PCA-reduced CLIP (Radford et al., 2021) embedding, and $\alpha^j \in \mathbb{R}$ is an opacity value that defines the existence of a specific bounding box. [The maximum token count is fixed to $J = 30$ between training and inference for model stability. During training, opacity is binary: 1 for present objects and 0 for absent ones. At inference, we apply a fixed threshold of 0.5 to decide whether a predicted object is included or discarded. The model can produce layouts with $j < 30$ objects, because all tokens with opacity $< 0.5$ are discarded. As we show in Figure 18 in Appendix P, the model typically outputs opacity values close to 0 or 1. The final layouts would be practically unchanged if the threshold were perturbed.]

**Rectified Flow Preliminaries.** We briefly recap the basics of rectified flow introduced in (Liu et al., 2022). Let $I$ be a set of training sample indices and $\{\mathbf{x}_i\}_{i \in I}$ the ground-truth samples whose distribution we would like to learn using our model $v$. We linearly interpolate between Gaussian noise $\mathbf{x}_i(0)$ and samples $\mathbf{x}_i(1) \equiv \mathbf{x}_i$ across timesteps $t \in [0, 1]$ as follows:

$$\mathbf{x}_i(t) = (1 - t) \cdot \mathbf{x}_i(0) + t \cdot \mathbf{x}_i(1). \tag{2}$$

The model $v$ is trained to take $(\mathbf{x}_i(t), t)$ as input and to predict the derivative of the path between $\mathbf{x}_i(0)$ and $\mathbf{x}_i(1)$, which according to Eq. 2 is $\mathbf{x}_i(1) - \mathbf{x}_i(0)$. The training objective is:

$$\min_v \int_0^1 \mathbb{E}_i[||(\mathbf{x}_i(1) - \mathbf{x}_i(0)) - v(\mathbf{x}_i(t), t)||^2] dt \tag{3}$$

and is optimized with stochastic gradient descent. This optimization is carried out across all available samples of the ground-truth distribution. Following (Liu et al., 2022), noisy values $\mathbf{x}_i(0)$ are resampled at each epoch. The end result is a network $v$, which is effective at predicting the direction from a noisy sample at an intermediate timestep towards the target distribution. Since a single evaluation may be noisy, the inference is performed by integrating over $T$ timesteps:

$$\mathbf{x}_i(1) = \mathbf{x}_i(0) + \sum_{t=1}^{T} v(\mathbf{x}_i(\frac{t-1}{T}), \frac{t}{T}) \cdot \frac{1}{T}. \tag{4}$$

**Our Model Architecture.** Our rectified flow model is built from multihead AdaLN transformer blocks, which can process tokens $\{\mathbf{x}_i^j\}_{j \in J}$ to iteratively denoise them (Peebles & Xie, 2023). The timestep $t$, bounding box coordinates $\mathbf{b}_i^j(t)$, and opacity values $\alpha_i^j(t)$ are sinusoidally encoded. The timestep $t$ and a linear projection of the global $P_i$'s CLIP encoding are passed as conditioning of the adaptive layer normalization of the transformer blocks. The tokens represent the objects in the layout and are processed all at once.

Inference begins at $t = 0$ with the set of tokens $\{\mathbf{x}_i^j(t)\}_{j \in J} \equiv \{\mathbf{x}_i^j(0)\}_{j \in J}$ initialized from Gaussian noise. Our model then iteratively processes and updates the tokens from $t = 0$ to $t = 1$ over $T$ iterations using Eq. 4 based on the global prompt conditioning $P_i$. We project this output back to the dimension of $\mathbf{x}_i^j(t)$ before sinusoidal encoding, in order for the module to serve as the rate of change of $\mathbf{x}_i^j(t)$. A single inference step can be summarized as:

$$\{\mathbf{x}_i^j(t)\}_{j \in J} \leftarrow \{\mathbf{x}_i^j(t - \frac{1}{T})\}_{j \in J} + v(\{\mathbf{x}_i^j(t - \frac{1}{T})\}_{j \in J}, t - \frac{1}{T}, P_i) \cdot \frac{1}{T}, \tag{5}$$

Following Eq. 5 until $t = 1$ yields the final layout $\{\mathbf{x}_i^j(1)\}_{j \in J}$ that contains PCA-reduced CLIP embeddings, bounding boxes, and opacities. Tokens with $\alpha_i^j(1) < 0.5$ are considered unused and discarded, please see the supplement for further explanation. For image generation, we unproject each $\mathbf{c}_i^j(1)$ from the PCA space back into the CLIP feature space and pass the embeddings directly into the downstream image generation module.

For visualization of the layouts, we follow the common practice when interpreting visual representations in natural language (Kerr et al., 2023; Qin et al., 2024) and decode CLIP embeddings to text by comparing them to label queries from the user, and selecting the closest query in the embedding space. In the supplement, we explain the RePaint (Lugmayr et al., 2022; Schröppel et al., 2024) technique for rectified flow to enable *partial layout conditioning*. This enables our model to be guided by partial layouts where only some boxes or labels are given (see Fig. 6). We additionally show how we can impose inter-bounding box positional constraints (i.e., place *A* to the *left* of *B*) by adding a directional drift on the bounding boxes during inference. The ability to control our model through these conditions allows it to also work in concert with an LLM to handle complex prompts, where the role of the LLM is to extract the constraints from the prompt, and our method takes care of generating the remaining unspecified scene details.

**Training.** To construct a training sample from the ground-truth image layout $i$, we create $\mathbf{c}_i^j$ and $\mathbf{b}_i^j$ for each bounding box $j$, and initialize $\alpha_i^j$ to 1. To ensure a consistent amount of tokens, we pad the samples by adding tokens with $\alpha_i^j = 0$ and $\mathbf{b}_i^j = 0$, and $\mathbf{c}_i^j$ to the null string embedding. We now treat $\{\mathbf{x}_i^j\}_{j \in J} \equiv \{\mathbf{x}_i^j(1)\}_{j \in J}$, sample $\{\mathbf{x}_i^j(0)\}_{j \in J}$ from Gaussian noise, draw $t$ uniformly from $[0, 1]$, and compute the set of tokens $\{\mathbf{x}_i^j(t)\}_{j \in J}$ by adapting the formula from Eq. 2, which are then passed to the model as input. We refer to the output of the model as $v(\{\mathbf{x}_i^j(t)\}_{j \in J}, t, P_i)$ and compute the training loss derived from Eq. 3:

$$\mathcal{L} = \sum_{i \in I, j \in J} ||\mathbf{x}_i^j(1) - \mathbf{x}_i^j(0) - v(\{\mathbf{x}_i^j(t)\}_{j \in J}, t, P_i)_j||^2. \tag{6}$$

**Human Evaluation.** Given the novelty of our problem domain, we argue that human evaluation is most reliable for assessing the plausibility and variety of layouts and therefore introduce a human-evaluation study which can be repeated by others. Assessing human opinions for these criteria directly on layouts is challenging: the evaluators require time to understand the layout diagrams and explain them, and furthermore, assessments are hard to make without actually seeing the image. Following the design principles presented by Otani et al. (2023) in their work on human evaluation of text-to-image generation: 1) *the (evaluation) task should be simple*, and 2) *results should be interpretable*. Therefore, we show participants only images, and omit the underlying image layouts entirely, which may take effort to understand. To make the results interpretable, participants rate these images for their *plausibility* and *variety* on a Likert scale (as recommended in Otani et al. (2023)) from 1 to 5. Image qualities that are assessed in other studies (for example, the overall quality and aesthetic appeal of the image in Liang et al. (2024)) are highly dependent on the conditioned image generator. Therefore, we consider these misleading for our case.

The study is approved by the Ethics Review Board of our institution and complies with local wage regulations. To keep the cost of a survey below 250 USD, we survey 60 participants, who each assess four text-to-layout generation methods at once, each providing ten plausibility questions and ten variety ratings. To increase the stability of the results and test on a larger sample set, each rating is for a collection of three images from the same prompt. The subset of collections, as well as the order they are displayed to the participant, are randomized to control for any potential effects of a fixed ordering. An expanded explanation of our survey

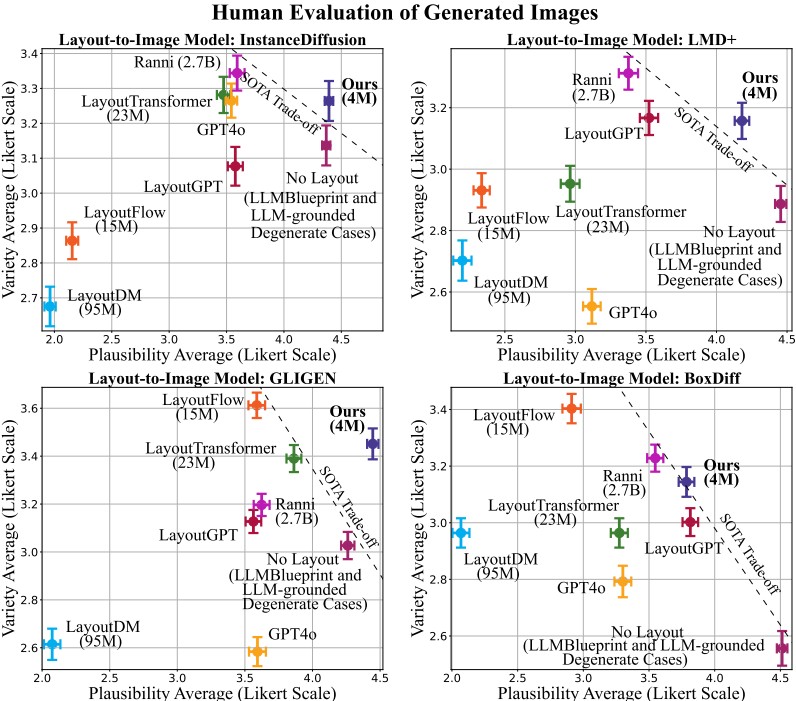

Figure 4: **Human Survey Results.** Our method offers an equal or superior trade-off between plausibility and variety across all measured layout-to-image generators, while being a much smaller model. The error bars indicate standard error.

design, including the text instructions and screenshots of the survey, can be found in the supplemental material.

## 4 Experiments

**Dataset.** We test our method's ability to learn a variety of plausible scene layouts by both training and evaluating on the full ADE20K dataset (Zhou et al., 2018), which contains approximately 27K images and ground-truth layouts for indoor and outdoor scenes, and a rich collection of object arrangements. [ We use the scene labels from the Scene UNderstanding (SUN) dataset Xiao et al. (2010), (on which ADE20K is based), as prompts for both training and evaluation. These labels are a single noun, such as 'street' or 'airport_terminal'. Labels are minimally preprocessed by replacing underscores with spaces. So, a sample with the SUN scene label 'street' will have the caption text 'street', and a sample with the SUN scene label 'airport_terminal' would have the caption text 'airport terminal'.]

We use the top 30 largest bounding boxes per sample, as this is the default maximum number of bounding boxes supported by InstanceDiffusion (Wang et al., 2024) and we pad samples with fewer bounding boxes. For evaluation, we use the 15 highest represented categories and add in five randomly selected categories to include the dataset's long tail distribution. For each evaluated model, we generate 30 layouts for all 20 selected prompts, and an image conditioned on each layout and corresponding global prompt. The size of this collection of images makes it feasible to assess the results with human evaluation.

**Baselines.** We compare our method against prior works which are capable of unconstrained layout generation. For LLM-baselines, we evaluate against LayoutGPT (Feng et al., 2023) and Ranni (Feng et al., 2024), but discard LLM-grounded Diffusion (Lian et al., 2024) and LLM Blueprint (Gani et al., 2024), as these give degenerate cases in 90%+ of measured cases in our domain (see Fig. 2). To see if LLM performance can be improved with proper in-context examples, we adapt the template from (Lian et al., 2024) with relevant in-context-learning examples from ADE20K. For the underlying LLM, we select the large-scale LLM

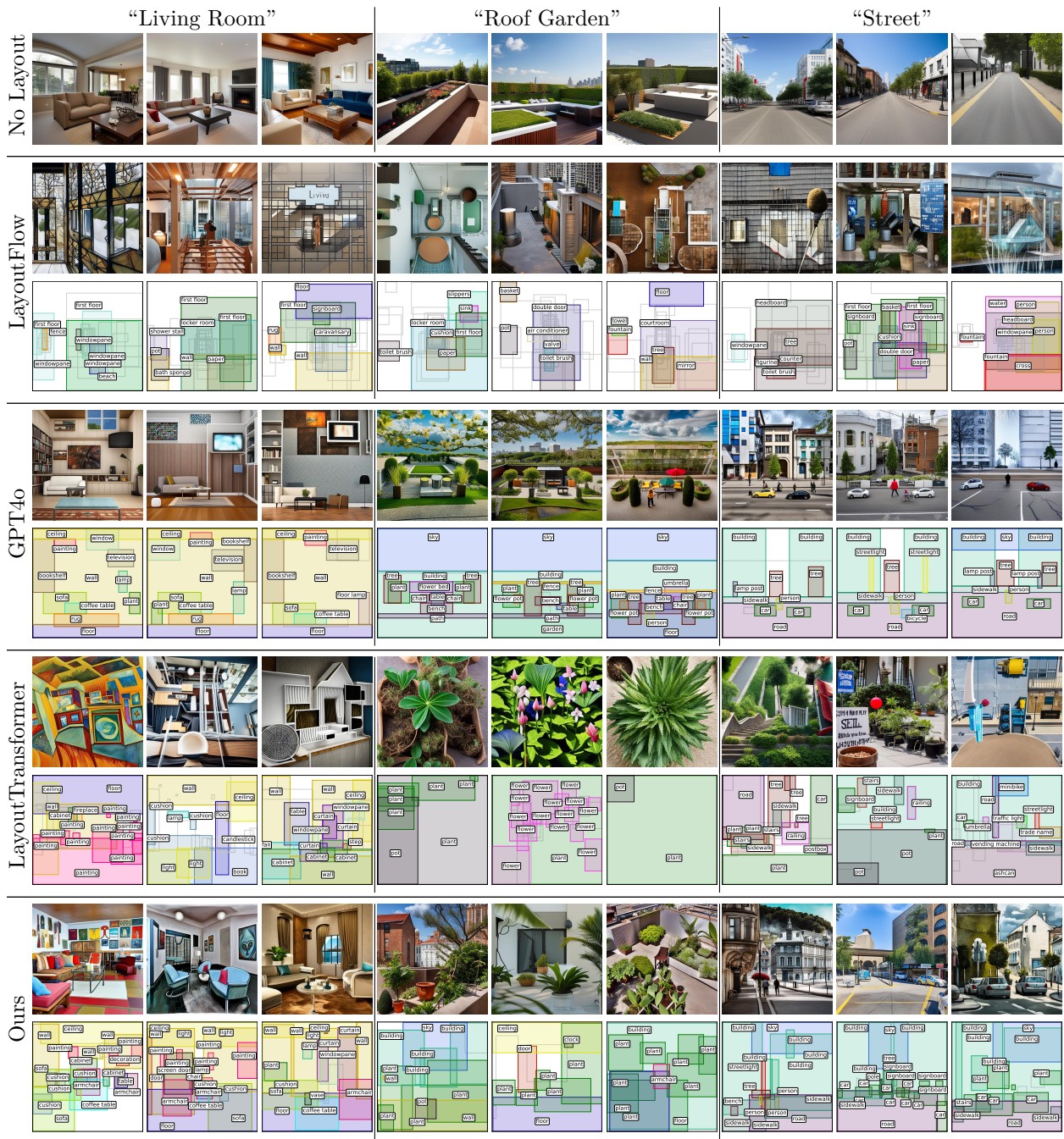

Figure 5: **Qualitative comparison** (Best viewed up close). Layout objects that are depicted in the generated image are highlighted and labeled. From a visual inspection, having no layout produces scenes of little variation in content. LayoutFlow's layouts do not appear to capture scene structure. GPT4o's layouts lack variety. Layout Transformer produces layouts with implausible arrangements of objects, leading to images which do not depict the global prompt accurately. Our method creates plausible and varied layouts, leading to images that are diverse and look realistic. These observations are supported by our human evaluation in Fig. 4. Zoomed-in versions of these layouts for printing are available in the supplemental.

GPT4o (OpenAI et al., 2024), and refer to this baseline simply as GPT4o. The full template is in the supplement. [For prompt rewriting, we utilize Prompt Enhancer Wang et al. (2025), a strong, recent approach

| Model | CMMD ($\downarrow$) | FID ($\downarrow$) | KID ($10^{-2}$) ($\downarrow$) | HSPv2 ($\uparrow$) | Image Reward ($\uparrow$) | VQA ($\uparrow$) |
|---|---|---|---|---|---|---|
| LayoutFlow | 0.25 ±0.01 | 0.80 ±0.12 | 0.88 ±0.07 | 0.2257 ±0.0003 | -1.01 ±0.01 | 0.803 ±0.002 |
| LayoutDiffusion | 0.40 ±0.01 | 1.08 ±0.12 | 1.99 ±0.07 | 0.1881 ±0.0003 | -2.11 ±0.01 | 0.342 ±0.002 |
| LayoutTransformer | 0.06 ±0.01 | 0.44 ±0.12 | 0.30 ±0.07 | 0.2269 ±0.0003 | -1.00 ±0.01 | 0.746 ±0.002 |
| GPT4o | 0.09 ±0.01 | 0.94 ±0.12 | 0.45 ±0.07 | 0.2510 ±0.0003 | -0.51 ±0.01 | 0.880 ±0.002 |
| Ranni | 0.07 ±0.01 | 0.71 ±0.12 | 0.30 ±0.07 | 0.2517 ±0.0003 | -0.34 ±0.01 | 0.896 ±0.002 |
| LayoutGPT | 0.29 ±0.01 | 2.83 ±0.12 | 1.76 ±0.07 | 0.2515 ±0.0003 | -0.26 ±0.01 | **0.928** ±0.002 |
| **[Prompt Enhanced Models]** | | | | | | |
| LLMBlueprint | 0.21 ±0.01 | 1.18 ±0.12 | 0.71 ±0.07 | 0.2527 ±0.0003 | -0.29 ±0.01 | 0.903 ±0.002 |
| GPT4o | 0.17 ±0.01 | 1.29 ±0.12 | 0.87 ±0.07 | 0.2489 ±0.0003 | -0.45 ±0.01 | 0.889 ±0.002 |
| LayoutGPT | 0.22 ±0.01 | 0.98 ±0.12 | 0.74 ±0.07 | 0.2469 ±0.0003 | -0.16 ±0.01 | 0.905 ±0.002 |
| LLMGroundedDiffusion | 0.19 ±0.01 | 1.08 ±0.12 | 0.76 ±0.07 | 0.2534 ±0.0003 | -0.28 ±0.01 | 0.916 ±0.002 |
| Ranni | 0.20 ±0.01 | 1.14 ±0.12 | 0.56 ±0.07 | 0.2501 ±0.0003 | -0.16 ±0.01 | 0.897 ±0.002 |
| **[Fine-Tuning]** | | | | | | |
| Ranni | 0.21 ±0.01 | 0.80 ±0.12 | 0.67 ±0.07 | **0.2591** ±0.0003 | **0.09** ±0.01 | 0.903 ±0.002 |
| Ours | **0.03** ±0.01 | **0.17** ±0.12 | **0.16** ±0.07 | 0.2477 ±0.0003 | -0.32 ±0.01 | 0.884 ±0.002 |

Table 1: **Image Metrics Comparison.** We evaluate traditional metrics and compare the images generated from layouts of different layout generators. To avoid biases of the image generator, we show the best score among the layout-to-image generators InstanceDiffusion (Wang et al., 2024), GLIGEN (Li et al., 2023), BoxDiff (Xie et al., 2023), and LMD+ (Lian et al., 2024) for each layout generator. [We define effective-difference thresholds for generated image metrics by computing the mean absolute difference between metric evaluations on $n = 10$ sample sets drawn from the same distribution. Differences in the metric within this range are not considered meaningfully different. The metrics are computed we follow the sampling and evaluation procedure described in Section 4, paragraph: Generated Image Metrics. We sample ADE20K ground-truth images from 20 categories, (30 samples per category) to form the generated distribution. For the CMMD, FID, and KID metrics, we use images generated from BoxDiff, conditioned on ground-truth ADE20K layouts as the reference distribution, using the same categories and sampling scheme. Thresholds are included in the table with the plus/minus margins] Our method achieves strong or state-of-the-art numbers for measured metrics. Although their metrics are strong, Ranni and LayoutGPT are susceptible to degenerate solutions (see Fig. 2) [Our method outperforms all baselines on CMMD, FID, and KID, indicating better alignment with the training distribution. Although expanding the prompt generally improves baseline metrics over their default implementations, the main advantage of our method—accurately learning the full target distribution—remains.] [See Appendix Q for more analysis on the fine-tuned Ranni model performance]

(CVPR 2026) that expands under-constrained prompts into varied text descriptions. PromptEnhancer does not natively support layout generation, so we feed the extended prompts into our LLM baselines and report the results.] [For a fine-tuning baseline, following Ranni's training procedure, we fine-tune LLaMA 2–7B Chat et al. (2023) using LoRA Hu et al. (2021) on both element- and box-level representations. We convert ADE20K into Ranni's prompt–response format and adopt the reported hyperparameters (10K iterations, effective batch size 64).]

We test against the UI generators LayoutTransformer (Gupta et al., 2021), LayoutDM (Inoue et al., 2023) and LayoutFlow (Guerreiro et al., 2024) by treating the global caption as a scene-wide bounding box and conditioning the model on this bounding box during inference. When training models, we stuck to their respective provided training settings.

**Human Evaluation.** As shown in Fig. 4, our model achieves a state-of-the-art balance in image plausibility and variety across all measured layout-to-image generators: InstanceDiffusion (Wang et al., 2024), GLIGEN (Li et al., 2023), BoxDiff (Xie et al., 2023), and LMD+ (Lian et al., 2024). The error bars indicate standard error ($s = \frac{\sigma}{\sqrt{n}}$) of the mean human rating, calculated using `numpy` We assume normally distributed errors. display the approximate number of model parameters added to the full text-to-layout-to-image pipeline by the layout generators that can be locally run. Our model is the smallest by over a factor 3.

**Visual Results.** We provide a qualitative overview of the generated layouts and the final images in Fig. 5, with InstanceDiffusion (Wang et al., 2024) as the layout-to-image model. We label bounding boxes by querying with all text labels present within ADE20K. From a visual inspection, LayoutTransformer struggles with arranging objects in spatially plausible manner. GPT4o layouts appear somewhat flat, while struggling to make a variety of layouts. Our method appears to produce both plausible and diverse images across a range of global prompts of indoor and outdoor settings.

**Generated Image Metrics.** We compute established image generation metrics CMMD (Jayasumana et al., 2024), FID (Heusel et al., 2018), KID (Bińkowski et al., 2021), VQA (Lin et al., 2024), HPSv2 (Wu et al., 2023a), and ImageReward (Xu et al., 2023). CMMD, FID and KID compare the distribution of generated images with a ground-truth distribution, while VQA, HSPv2 and ImageReward assess general image quality and alignment with a global caption. [To evaluate a layout generation method, we generate layouts across 20 categories of ADE20K, 30 samples per category (600 layouts total). We generate one image per layout, and compute metrics on this set of 600 images. The reference distribution for CMMD, FID, KID is constructed analogously. We sample ground-truth ADE20K layouts from the same 20 categories, with 30 samples per category, generate one image per layout, and the resulting 600 image set is our reference distribution. HSPv2, VQA, and ImageReward are computed directly on the generated images and do not require a reference distribution.] For each layout generator, we display the optimal score over the possible combinations of layout and image generator ((Wang et al., 2024; Li et al., 2023; Xie et al., 2023; Lian et al., 2024)). Images from degenerate layouts from Ranni and Layout-GPT are discarded to more clearly assess the layout's influence. The results are shown in Tab. 1, with state-of-the-art performance in CMMD, FID, KID and HSPv2, and strong results in ImageReward and VQA.

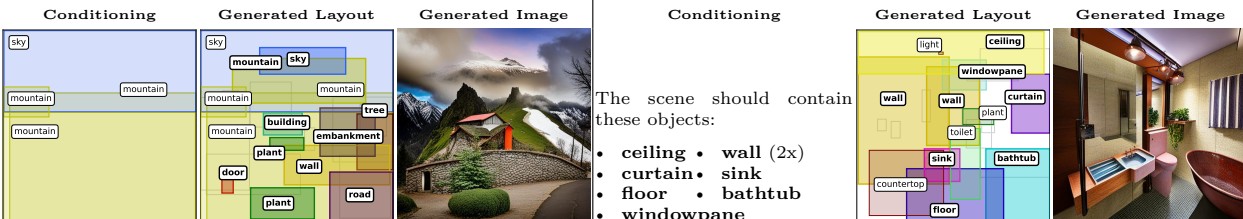

Figure 6: **Disentangled Generation.** Disentangled generation for scenes with the prompt *Snowy Mountain* with a partial layout (**Left**), and *Bathroom* with a *bag of words* (**Right**).

**Scene Layout Metrics, and Speed.** We consider how to best assess scene layouts for unconstrained prompts. The traditional UI generation metrics of Alignment (Lee et al., 2020) and Overlap (Li et al., 2019) scores are not salient, as real world images often have misaligned or overlapping bounding boxes. Likewise, the layout-FID (Heusel et al., 2018) metric requires a layout-GAN discriminator to compute, which we do not have in this new domain. We compute a standard mIoU (Kikuchi et al., 2021) averaged across sampled scene categories. To provide a more complete evaluation, we introduce metrics aimed to quantify a generated layout's *plausibilty* and *variety* that we describe in full in the supplementary material. We measure the model's generation time on batches of 30 layout samples on an Nvidia A6000 GPU with 32 AMD Ryzen 9 5950X CPUs, 125 GB RAM, except for GPT4o that is accessed through an API. Numerical results are provided in the supplement. Notably, we achieve the highest performance in positional likelihood (how plausibly objects are arranged) and mIoU. Our method ranks second in speed only to LayoutFlow, but we observe no definitive improvement in its layout statistics when the number of inference steps are raised to match our model's speed.

**Additional Model Features.** We briefly highlight qualities of SLayR which make it appealing to use: In Fig. 6, we show examples of our model's performance in different partial layout generation settings. This feature gives users even more fine-grained control over the image generation process. Additionally, we demonstrate how a text-to-layout-to-image pipeline allows for editing of generated images in Fig. 7. This is accomplished through modifying the intermediate scene layout, and rerunning layout-to-image generator with the original seed and global prompt.

## 5 Conclusion

We have introduced a text-to-layout model, incorporating it into a text-to-image pipeline with an intermediate and controllable layout representation. With a substantially smaller model, we can generate images with a start-of-the-art balance in plausibility and variety, while achieving high or state-of-the-art performance in generated image quality metrics among competing baselines. In addition, we have introduced a suite of

| User Action | Image Layout | Image | User Action | Image Layout | Image |
|---|---|---|---|---|---|
| A layout for a "Conference Room" with a "plant" bounding box guides the image generation. | | | The "plant" is moved. The plant is moved in generated image. | | |
| We remove the "plant". The plant disappears in the generated image. | | | We replace "plant" with "painting". The generated image now contains a painting instead of a plant. | | |

Figure 7: **Editing.** We show how our pipeline enables user editing of images by altering the intermediate scene layout representation. Individual objects can be easily moved, removed, and replaced.

metrics for the new task of scene layout generation, with which we established the foundation to explore image generation pipelines with explicit intermediate layouts in the future. [Our current experiments are conducted on ADE20K, and therefore focus on indoor and outdoor scene categories present in that dataset. This is a property of the training data rather than a limitation of the method itself. The proposed approach is not inherently restricted to these domains and can be extended to broader, open-domain settings given appropriate training data. For example, an open-domain layout dataset could be constructed by applying object detection models (e.g., YOLO Sapkota & Karkee (2026)) to large-scale image collections such as LAION Schuhmann et al. (2021), and converting the outputs into layout annotations. Thus, while our evaluation is in-domain, the method is general and can scale to more diverse scene distributions with suitable data.]

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
