# OpenReview forum: "Scene Layout Generation with Rectified Flow"
_TMLR — Decision pending for TMLR_

### Review · Reviewer_CU4f · 2026-03-02

**Summary Of Contributions:**

This paper studies text-to-layout generation for open-set scenes, where prompts are coarse and ambiguous (e.g., “beach”, “street”) and the goal is to produce layouts that are both plausible and diverse.
This paper proposes a rectified-flow method with DiT to generate the layout, called SLayR. It generates a fixed-length set of object slots. Each slot includes a bounding box, a continuous open-vocabulary semantic embedding (implemented via PCA-reduced CLIP features), and a opacity gate to model variable object counts.
The proposed text-to-layout generator is integrated with off-the-shelf layout-to-image models to form a complete text-to-layout-to-image pipeline.
The paper also introduces an human evaluation method for plausibility–variety trade-offs.

Strengths:
1. Clear problem settings. This paper targets under-explored “unconstrained prompt” for text-to-layout generation ; (iii)
2. The representation in the proposed method is practical and interpretable for open-vocabulary semantics.
3. The method in this paper provide an end-to-end pipeline validation and the human evaluation protocol is adapted to the task in this paper.

Weaknesses:
1. The compared baselines are somewhat weak. The LLM-centric baseline (GPT-4o) only uses a strong LLM with chain-of-thought reasoning, without incorporating external knowledge such as retrieval-backed priors, which may make it an inadequate LLM-centric baseline.
2. The setting is effectively in-domain rather than truly open-domain, making it applicable only to indoor/outdoor scenes.
3. The definition of ground truth and degeneration rates in the experiments is unclear.

**Audience:**

Yes

**Audience Explanation:**

1. Researchers and practitioners working on text-to-image generation, controllable generation, intermediate representations (layout/scene graphs), and diffusion/flow-based generative modeling would likely find this paper relevant.

2. The focus on unconstrained prompts highlights a practical failure mode of existing LLM-based layout approaches.  The proposed open-vocabulary layout representation and repeatable plausibility–variety evaluation protocol can be useful for future work on controllable and interpretable text-to-image pipelines.

3. The results may also interest those building production systems that require generating diverse yet plausible scene structures from underspecified prompts.

**Claims And Evidence:**

Yes

**Claims Explanation:**

1. The experimental results show improved image-distribution metrics (e.g., FID/KID) across multiple off-the-shelf layout-to-image generators, indicating that the reported gains are not renderer-specific.

2. Quantitative analyses show a much lower degenerate-layout rate on unconstrained prompts compared to LLM-based baselines, directly supporting the paper’s core motivation.

3. Extensive qualitative comparisons across different models (layout-to-image models) and generation stages provide additional evidence for the plausibility and diversity of the proposed approach.

**Requested Changes:**

1. The paper compares against LLM-centric baselines, but it is unclear whether these baselines incorporate a stronger LLM+knowledge pipeline (e.g., retrieval-backed or statistics-backed scene priors, and N-sample generation with plausibility/diversity-aware reranking). Since the design of baselines can significantly affect the variety–plausibility trade-off in unconstrained settings, I suggest the authors clarify the exact baseline pipeline, and consider including such a stronger baseline or justify why it is out of scope.

2. The prompts are highly consistent with the ADE20K training distribution (i.e., they correspond to ADE20K categories), so strictly speaking this is “in-domain” rather than truly open-domain. ADE20K contains real layout annotations for indoor and outdoor scenes with rich object arrangements. Does this imply that the proposed method is limited to generating indoor/outdoor scenes?

3. For the calculation of degeneration rates, such as Figure 2 ("99%/90%/51%/25%/0%"), are multiple samples generated? How is the ground-truth determined? This is a critical question. The paper appears to use objects from ADE20K in-context examples as ground truth, which results in a 0% degeneration rate for the proposed method. The rationale and validity of this approach, as well as the interpretation of a 0% rate, may need further explanation.

4. Writing Issue: The bottom horizontal line is missing in Table 1.

---

> ### Author Response · Authors · 2026-05-12
> **Reviewer CU4f Requested Change 1**
>
> Requested Change: The paper compares against LLM-
> centric baselines, but it is unclear whether these baselines incorporate a stronger
> LLM+knowledge pipeline (e.g., retrieval-backed or statistics-backed scene priors, and N-sample generation with plausibility/diversity-aware reranking). Since
> the design of baselines can significantly affect the variety–plausibility trade-off in
> unconstrained settings, I suggest the authors clarify the exact baseline pipeline,
> and consider including such a stronger baseline or justify why it is out of scope.
>
> Answer: First, to clarify details of LLM-baselines introduced in Section 4, paragraph
> Baselines: LayoutGPT, LLM-Grounded Diffusion, and LLM-Blueprint provide
> the LLM the user prompt inside a template with in-context layout construction
> examples. Ranni finetunes the LLM on the layout generation task, in addition
> to using a template with in-context examples during inference. Our constructed
> baseline, referred to as GPT4o in the main text, uses a custom template with
> ADE20K-specific in-context examples and encourage chain-of-thought reasoning
> during inference. The full template can be found in Figure 12 in Appendix L.
>
> Regarding potential baseline extensions:
>
> N-sample generation or plausibility/diversity reranking: we argue that LLM
> candidates already consume significantly more time and compute resources than
> our proposed method (See Table 2 in Appendix C). While these techniques might
> refine results, to assess the models’ capabilities for fast and efficient generation,
> we find it most fair to compare all methods by evaluating their entire unrefined
> output distribution without complex engineered systems around them.
> Retrieval backed/statistics backed priors: Adding RAG [3] back-end calls
> will further increase LLM method’s time and compute resources. We seek to
> fairly assess of the models’ fast and efficient generation, and argue that RAG
> is out of scope. However, we do investigate directly instilling ADE20K-domain
> specific knowledge into the LLM using a fine-tuning experiment suggested by
> the other reviewers. The method and results are as follows:
>
> We implement this suggestion by fine-tuning the Ranni baseline, which already builds on LayoutGPT and whose methodology already leverages LLM
> fine-tuning (unlike the original LayoutGPT). Following Ranni’s training procedure, we fine-tune LLaMA 2–7B Chat [1]
> using LoRA [2] on both element- and box-level representations. We convert
> ADE20K into Ranni’s prompt–response format and adopt the reported hyper-
> parameters (10K iterations, effective batch size 64).
> We compare this LLM-based baseline to our method using standard image
> quality metrics:
>
> Model  |  CMMD ($\downarrow$)  |  FID ($\downarrow$)  |  KID ($10^{-2}$) ($\downarrow$)  |  HSPv2 ($\uparrow$)  |  Image Reward ($\uparrow$)  |  VQA ($\uparrow$)
> | :--- | ---: | ---: | ---: | ---: | ---: | ---: |
> Ranni (fine-tuned)  | 0.21  | 0.80  | 0.67  | 0.26  | 0.09  | 0.90 |
> Ours  | 0.03  |  0.17  | 0.16  | 0.25  | -0.32 | 0.88 |
>
> Our method substantially improves CMMD, FID, and KID, indicating better
> alignment with the training distribution than the LLM baseline, even if HSPv2,
> ImageReward and VQA measures slightly prefer the LLM generations.
>
> For further analysis of these results, see our address of reviewer 5wqL's first requested change.
>
> [1] Hugo Touvron et al. Llama 2: Open foundation and fine-tuned chat models,
> 2023.
>
> [2] Edward J. Hu, Yelong Shen, Phillip Wallis, Zeyuan Allen-Zhu, Yuanzhi Li,
> Shean Wang, Lu Wang, and Weizhu Chen. Lora: Low-rank adaptation of
> large language models, 2021.
>
> [3] Patrick Lewis, Ethan Perez, Aleksandra Piktus, Fabio Petroni, Vladimir
> Karpukhin, Naman Goyal, Heinrich K¨uttler, Mike Lewis, Wen tau Yih, Tim
> Rockt¨aschel, Sebastian Riedel, and Douwe Kiela. Retrieval-augmented gen-
> eration for knowledge-intensive nlp tasks, 2021.

---

> ### Author Response · Authors · 2026-05-12
> **Reviewer CU4f Requested Change 2**
>
> Requested Change: The prompts are highly consistent with the ADE20K
> training distribution (i.e., they correspond to ADE20K categories), so strictly
> speaking this is “in-domain” rather than truly open-domain. ADE20K contains
> real layout annotations for indoor and outdoor scenes with rich object arrangements. Does this imply that the proposed method is limited to generating
> indoor/outdoor scenes?
>
> Answer: Our current experiments are conducted on ADE20K, and therefore
> focus on indoor and outdoor scene categories present in that dataset. This is
> a property of the training data rather than a limitation of the method itself.
> The proposed approach is not inherently restricted to these domains and can
> be extended to broader, open-domain settings given appropriate training data.
>
> For example, an open-domain layout dataset could be constructed by applying
> object detection models (e.g., YOLO [4]) to large-scale image collections such
> as LAION [5], and converting the outputs into layout annotations. Thus, while
> our evaluation is in-domain, the method is general and can scale to more diverse
> scene distributions with suitable data.
>
> [4] Ranjan Sapkota and Manoj Karkee. Ultralytics yolo evolution: An overview
> of yolo26, yolo11, yolov8 and yolov5 object detectors for computer vision
> and pattern recognition, 2026.
>
> [5] Christoph Schuhmann, Richard Vencu, Romain Beaumont, Robert Kacz-
> marczyk, Clayton Mullis, Aarush Katta, Theo Coombes, Jenia Jitsev, and
> Aran Komatsuzaki. Laion-400m: Open dataset of clip-filtered 400 million
> image-text pairs, 2021.

---

> ### Author Response · Authors · 2026-05-12
> **Reviewer CU4f Requested Change 3**
>
> Requested Change: For the calculation of degeneration rates, such as
> Figure 2 (”99%/90%/51%/25%/0%”), are multiple samples generated? How is
> the ground-truth determined? This is a critical question. The paper appears to
> use objects from ADE20K in-context examples as ground truth, which results
> in a 0% degeneration rate for the proposed method. The rationale and validity
> of this approach, as well as the interpretation of a 0% rate, may need further
> explanation.
>
> Answer: We generate layouts in Figure 2, by using the methodology in Sec
> 4, paragraph Dataset: we query each baseline on 20 scene category labels (e.g.,
> ”street”, ”roof garden”) present in ADE20K, collecting 30 sample layouts per
> category (600 samples total), i.e, multiple samples are generated.
> We define a layout as degenerate if it (i) contains no bounding boxes, or
> (ii) contains a single bounding box that occupies the entire frame, with the
> scene category name as its label. These layouts are degenerate in the sense that
> they do not provide any meaningful scene structure information to the layout-
> to-image generator, making text-to-layout generator superfluous to generating
> the image. Degeneration rates do not rely on ADE20K annotations as ground
> truth. The criterion is based solely on the structure of the generated layout.
> Under this definition our method achieves a 0% degeneration rate: none of
> the 600 generated layouts are degenerate (each contains at least 2 bounding
> boxes).
>
> Regarding the ADE20K in-context examples: the high degeneration rates
> of off-the-shelf LLM methods (shown in Figure 2) motivates us to construct a
> stronger LLM-baseline with in-context examples of ADE20K layouts and chain-
> of-thought reasoning, referred to as the ’GPT4o’ baseline in the text. This
> reduces degeneration to 0%. However, as shown in Table 1, even with this
> prompt engineering, LLM-based methods produce layouts that are less aligned
> with the target distribution than our flow-based approach.

---

> ### Author Response · Authors · 2026-05-12
> **Reviewer CU4f Requested Change 4**
>
> Requested Change: Writing Issue: The bottom horizontal line is missing
> in Table 1.
>
> Answer: Thank you for pointing this out. We correct this in our manuscript.
> We will add the additional results, discussions, and clarifications to the document.

---

### Review · Reviewer_mYfs · 2026-04-28

**Summary Of Contributions:**

The paper proposes SLayR, a rectified-flow Transformer for text-to-layout generation. Given a broad scene-level prompt such as “beach,” “park,” or “lecture room,” the model generates a set of object-level layout tokens containing bounding boxes, CLIP-based object embeddings, and opacity values. These layouts are then used as intermediate conditions for existing layout-to-image generators, forming a text-to-layout-to-image pipeline.

The main contributions are: (1) formulating unconstrained scene-prompt layout generation as a rectified-flow modeling problem; (2) introducing a layout representation that combines object boxes, semantic embeddings, and existence/opacity scores; (3) evaluating the generated layouts after rendering them through multiple layout-to-image backends; and (4) demonstrating additional controllability through partial layout conditioning and layout editing.

The intermediate layout representation is potentially useful for interpretability and user control. However, I have significant concerns about the paper. I found the motivation not very compelling, since the proposed solution decomposes unconstrained text-to-image generation into a separate text-to-layout stage, but the paper does not clearly justify why this decomposition is preferable to simpler alternatives such as modern LLM-based prompt rewriting. I also found the paper difficult to follow in several important places, especially around the dataset construction, inference-time object selection, and metric computation. Overall, the idea may be useful, but the current submission does not make a sufficiently clear or convincing case.

**Audience:**

Yes

**Audience Explanation:**

Some TMLR readers may be interested in the paper because it studies structured intermediate representations for generative modeling. The text-to-layout-to-image formulation is relevant to researchers working on controllable generation, interpretable image synthesis, layout-conditioned diffusion models, and structured scene representation.

That said, I expect the interest to be limited unless the paper more clearly establishes the advantage of the proposed layout-based decomposition over simpler prompt-expansion or end-to-end generation baselines.

**Broader Impact Concerns:**

I do not see major broader-impact concerns unique to this paper beyond those generally associated with text-to-image generation systems. However, the paper should acknowledge that SLayR may inherit biases from ADE20K, CLIP embeddings, and the downstream layout-to-image generators.

**Claims And Evidence:**

No

**Claims Explanation:**

A central claim is that SLayR provides a better solution for unconstrained scene-prompt generation, but the paper does not evaluate a direct and natural alternative: using a modern LLM to expand a short prompt into a richer scene description before feeding it to a strong text-to-image model. Since the stated problem is that prompts such as “a beach” or “a park” are underspecified, prompt rewriting or scene expansion is a highly relevant baseline. Without this comparison, it is difficult to determine whether the proposed text-to-layout model is necessary.

The baseline set is also not convincing enough for the strength of the claims. Several compared methods are either older LLM-layout methods or adapted UI-layout generators. The UI-layout models are from a substantially different domain, where labels are usually closed-set and layouts follow different structural patterns. These baselines are useful for diagnosis, but they are not sufficient to establish state-of-the-art performance for open-world scene generation.

Some key experimental details are not clear enough to verify the claims. The paper does not clearly specify how the effective number of generated objects is determined at inference time, what text field is used as the ADE20K “caption,” or what exact image sets are used as references for CMMD, FID, KID, HPSv2, VQA, and ImageReward. These ambiguities make the empirical evidence difficult to assess

**Requested Changes:**

- Add a modern LLM prompt-rewriting / scene-expansion baseline.

Compare against a pipeline where a strong LLM expands prompts such as “a beach” or “a park” into detailed scene descriptions, which are then passed to a strong text-to-image model. This is necessary to justify the need for a separate text-to-layout model. Include more current LLM-based structured scene-planning or layout-generation baselines, using carefully designed prompts and modern models. The current LLM baselines do not sufficiently represent the capabilities of recent LLMs.

- Clarify the ADE20K text conditioning.

Explicitly state what is used as the prompt during training and evaluation. ADE20K does not provide natural free-form captions in the usual sense; it provides scene/category annotations. The paper should explain whether it uses the most specific scene label, a concatenation of labels, a cleaned category name, or another constructed prompt.

- Clarify inference-time object-count determination.

Explain how the model decides how many objects are present in the final layout. If this is controlled by the opacity value and a fixed threshold, the paper should report sensitivity to the opacity threshold and the maximum token count J.


- Clarify the metric computation protocol.

Specify the exact image sets and reference distributions used for CMMD, FID, KID, HPSv2, VQA, and ImageReward. In particular, distinguish clearly between evaluation against real ADE20K images and evaluation against images generated from ground-truth layouts.

---

> ### Author Response · Authors · 2026-05-12
> **Reviewer mYfs Requested Change 1**
>
> Requested Change: Add a modern LLM prompt-rewriting / scene-
> expansion baseline. Compare against a pipeline where a strong LLM expands prompts such as
> “a beach” or “a park” into detailed scene descriptions, which are then passed
> to a strong text-to-image model. This is necessary to justify the need for a sep-
> arate text-to-layout model. Include more current LLM-based structured scene-
> planning or layout-generation baselines, using carefully designed prompts and
> modern models. The current LLM baselines do not sufficiently represent the
> capabilities of recent LLMs.
>
> Answer: For prompt rewriting, we utilize Prompt Enhancer [6], a strong,
> recent approach (CVPR 2026) that expands under-constrained prompts into
> varied text descriptions. PromptEnhancer does not natively support layout
> generation, so we feed the extended prompts into our LLM baselines and report
> the results.
>
> Model  |  CMMD ($\downarrow$)  |  FID ($\downarrow$)  |  KID ($10^{-2}$) ($\downarrow$)  |  HSPv2 ($\uparrow$)  |  Image Reward ($\uparrow$)  |  VQA ($\uparrow$) |
> | :--- | ---: | ---: | ---: | ---: | ---: | ---: |
> LLMBlueprint  | 0.21  | 1.18  | 0.71  | 0.25  | -0.29  | 0.90 |
> GPT4o with COT  | 0.17  | 1.29  | 0.87  | 0.25  | -0.45  | 0.89 |
> LayoutGPT  | 0.22  | 0.98  | 0.74  | 0.25  | -0.16  | 0.91 |
> LLMGroundedDiffusion  | 0.19  | 1.08  | 0.76  | 0.25  | -0.28  | 0.92 |
> Ranni  | 0.20  | 1.14  | 0.56  | 0.25  | -0.16  | 0.90 |
> Ours  | 0.03  | 0.17  | 0.16  | 0.25  | -0.32  | 0.88 |
>
> Our method outperforms all baselines on CMMD, FID, and KID, indicating better alignment with the training distribution. Although expanding the
> prompt generally improves baseline metrics over their default implementations,
> the main advantage of our method—accurately learning the full target distribution—remains.
>
> [6] Linqing Wang, Ximing Xing, Yiji Cheng, Zhiyuan Zhao, Donghao Li,
> Tiankai Hang, Jiale Tao, Qixun Wang, Ruihuang Li, Comi Chen, Xin Li,
> Mingrui Wu, Xinchi Deng, Shuyang Gu, Chunyu Wang, and Qinglin Lu.
> Promptenhancer: A simple approach to enhance text-to-image models via
> chain-of-thought prompt rewriting, 2025

---

> ### Author Response · Authors · 2026-05-12
> **Reviewer mYfs Requested Change 2**
>
> Requested Change: Clarify the ADE20K text conditioning. Explicitly state
> what is used as the prompt during training and evaluation. ADE20K does not
> provide natural free-form captions in the usual sense; it provides scene/category
> annotations. The paper should explain whether it uses the most specific scene
> label, a concatenation of labels, a cleaned category name, or another constructed
> prompt.
>
> Answer: We use the scene labels from the Scene UNderstanding (SUN)
> dataset [7], (on which ADE20K is based), as prompts for both training and
> evaluation. These labels are a single noun, such as ’street’ or ’airport terminal’.
> Labels are minimally preprocessed by replacing underscores with spaces. So, a
> sample with the SUN scene label ’street’ will have the caption text ’street’, and
> a sample with the SUN scene label ’airport terminal’ would have the caption
> text ’airport terminal'
>
>
> [7] Jianxiong Xiao, James Hays, Krista A Ehinger, Aude Oliva, and Antonio
> Torralba. Sun database: Large-scale scene recognition from abbey to zoo. In
> 2010 IEEE Computer Society Conference on Computer Vision and Pattern
> Recognition, pages 3485–3492. IEEE, 2010

---

> ### Author Response · Authors · 2026-05-12
> **Reviewer mYfs Requested Change 3**
>
> Requested Change: Clarify inference-time object-count determination. Ex-
> plain how the model decides how many objects are present in the final layout.
> If this is controlled by the opacity value and a fixed threshold, the paper should
> report sensitivity to the opacity threshold and the maximum token count J.
>
> Answer: Each object token vector has a scalar entry called its opacity,
> which determines whether an object is present in the layout, or discarded. The
> maximum token count is fixed to J = 30 between training and inference for
> model stability. During training, opacity is binary: 1 for present objects and
> 0 for absent ones. At inference, we apply a fixed threshold of 0.5 to decide
> whether a predicted object is included or discarded. The model can produce
> layouts with j < 30 objects, because all tokens with opacity < 0.5 are discarded.
> See Section 3, paragraph: Our Model Architecture for the original discussion of
> this topic. As we show in Figure 18 in Appendix P, the model typically outputs
> opacity values close to 0 or 1. The final layouts would be practically unchanged
> if the threshold were perturbed.

---

> ### Author Response · Authors · 2026-05-12
> **Reviewer mYfs Requested Change 4**
>
> Requested Change: Clarify the metric computation protocol. Specify the
> exact image sets and reference distributions used for CMMD, FID, KID, HPSv2,
> VQA, and ImageReward. In particular, distinguish clearly between evaluation
> against real ADE20K images and evaluation against images generated from
> ground-truth layouts.
>
> Answer: We clarify the evaluation protocol described in Section 4, paragraph:
> Generated Image Metrics. To evaluate a layout generation method, we generate
> layouts across 20 categories of ADE20K, 30 samples per category (600 layouts
> total). We generate one image per layout, and compute metrics on this set of
> 600 images.
> The reference distribution for CMMD, FID, KID is constructed analogously.
> We sample ground-truth ADE20K layouts from the same 20 categories, with 30
> samples per category, generate one image per layout, and the resulting 600 image set is our reference distribution. HSPv2, VQA, and ImageReward are computed directly on the generated images and do not require a reference distribution. We
> do not use the ground-truth ADE20K images in any metric computation in Table 1.

---

### Review · Reviewer_5wqL · 2026-04-29

**Summary Of Contributions:**

This paper introduces SLayR, a compact transformer-based model that turns text prompts into detailed scene layouts. While LLM-based generators often struggle with vague prompts, SLayR uses rectified flow to build diverse and realistic spatial arrangements. With this intermediate step, users can precisely edit specific parts of the scene before the final layout is sent to a standard image generator. To accurately measure success in this novel, unconstrained setting, the authors contribute a comprehensive new benchmark suite that includes both numerical metrics and a rigorous, repeatable human evaluation procedure.

**Audience:**

Yes

**Audience Explanation:**

- This paper aims to address scene layout generation, a challenging research task with practical applications in layout planning and text-to-image generation.
- The proposed SLayR pipeline is model-agnostic, which can be integrated with advanced T2I models to achieve better performance.
- They provide comprehensive empirical results, human evaluations, and visualization cases.

**Claims And Evidence:**

Yes

**Claims Explanation:**

- The model creates a controllable pipeline. The denoised tokens represent bounding boxes, embeddings, and opacities, which provide a transparent mechanism for how the layout is formed. The intermediate representation also allows for human editing.
- The authors provide a direct visual and statistical comparison, demonstrating that existing LLM approaches often produce "degenerate layouts". SLayR’s ability to generate complex, multi-object layouts in these scenarios provides clear evidence of its superiority in handling ambiguity.
- Considering variety and plausibility, SLayR effectively breaks the trade-off where previous models had to sacrifice one for the other.

**Requested Changes:**

The layout generation process can be seen as a prompt enhancement over the original input prompt. Hence, there should be two missing baseline settings:
- Let the LLM derive the structure layout (like LayoutGPT but fine-tuned on the used data)
- Let the LLM do prompt rewriting to have a detailed description of the layout, and treat that description as the T2I prompt

Table 1 should include the threshold for what constitutes an effective difference for each metric, particularly as most HPS scores are around 0.25.

---

> ### Author Response · Authors · 2026-05-12
> **Reviewer 5wqL Requested Change 1**
>
> Requested Change: Let the LLM derive the structure layout (like Lay-
> outGPT but fine-tuned on the used data)
>
> Answer:
> We implement this suggestion by fine-tuning the Ranni baseline, which al-
> ready builds on LayoutGPT and whose methodology already leverages LLM
> fine-tuning (unlike the original LayoutGPT).
> Following Ranni’s training procedure, we fine-tune LLaMA 2–7B Chat [1]
> using LoRA [2] on both element- and box-level representations. We convert
> ADE20K into Ranni’s prompt–response format and adopt the reported hyper-
> parameters (10K iterations, effective batch size 64).
> We compare this LLM-based baseline to our method using standard image
> quality metrics:
>
> Model  |  CMMD ($\downarrow$)  |  FID ($\downarrow$)  |  KID ($10^{-2}$) ($\downarrow$)  |  HSPv2 ($\uparrow$)  |  Image Reward ($\uparrow$)  |  VQA ($\uparrow$)
> | :--- | ---: | ---: | ---: | ---: | ---: | ---: |
> Ranni (fine-tuned)  | 0.21  | 0.80  | 0.67  | 0.26  | 0.09  | 0.90 |
> Ours  | 0.03  |  0.17  | 0.16  | 0.25  | -0.32 | 0.88 |
>
> Our method substantially improves CMMD, FID, and KID, indicating better
> alignment with the training distribution than the LLM baseline, even if HSPv2,
> ImageReward and VQA measures slightly prefer the LLM generations.
>
> We also offer an anlysis below. The fine-tuned Ranni baseline struggles to produce consistently
> formatted layouts in text representation. Example model responses:
>
> Response 1:
>
> Here are the bounding boxes for each element in the lecture room:
>
> 1. Chalkboard: [0,0,1200,700]
>
> 2. Lectern: [100,0,600,200]
>
> 3. Students: *
>
> Student 1: [500,0,300,200]
>
> * Student 2: [700,0,300,200] * Student 3: [900,0,300,200]
>
> * Student 4: [1100,0,300,200]
>
> * Student 5: [1300,0,300,200]
>
> * Student 6: [1500,0,300,200]
>
> 4. Desk:
>
> * Desk 1: [0,500,1200,600]
>
> * Desk 2: [500,500,1200,600]
>
> * Desk 3: [1000,500,1200,600]
>
> * Desk 4: [1500,500,1200,600]
>
> 5. Chair:
>
> * Chair 1: [0,500,800,600]
>
> * Chair 2: [500,500,800,600]
>
> * Chair 3: [1000,500,800,600]
>
> * Chair 4: [1500,500,800,600]
>
> 6. Whiteboard: [1600,0,1200,700]
>
> Response 2:
>
> Based on the caption and elements provided, here are the bounding boxes for each element in the lecture room:
>
> Lecture Room: [0,0,10,10]
>
> Chalkboard: [3,3,6,4]
>
> Blackboard: [7,7,4,4]
>
> Whiteboard: [5,5,6,6]
>
> Desk: [8,8,2,3]
>
> Chair: [9,9,1,3]
>
> Student: [1,1,3,3]
>
> Professor: [6,6,4,4]
>
> Response 3:
>
> I imagine the image as a street scene with a horizontal orientation.
>
> Here are the bounding boxes for the elements ”street”:
> [x,y,w,h] = [0,0,300,50]
>
> This bounding box encloses the entire street scene in the image, including
> the road, the buildings, and any other elements present in the scene.
>
> Discussion:
>
> The assumed layout dimensions substantially vary, sometimes to be over
> 1K pixels wide/tall, or else shrinking the entire layout to be in a 10x10 frame.
> Thus, generated layouts are often ineffective conditioning on the downstream
> generated image.
> There are several reasons the model could be struggling: 1) Difference in
> generation domain: standard LLM-layout domains typically involve a few simple
> positional or numerical relationships described by the caption text (See Ranni
> Fig 2A, examples ’A white table with a vase, a pitcher, and a picture frame’,
> ’A teapot and two cups arranged on a table’), whereas our domain supplies no
> object nor relational cues, potentially limiting the model’s generation aptitude
> 2) Length of the training text: the high object counts (up to 30 objects) in
> ADE20K makes training text samples longer than the original Ranni (see again
> Fig 2A, examples ’A white table with a vase, a pitcher, and a picture frame’ has
> 4 bounding boxes, ’A teapot and two cups arranged on a table’ has 4 bounding
> boxes). The model could be struggling to scale to longer texts.
>
> [1] Hugo Touvron et al. Llama 2: Open foundation and fine-tuned chat models,
> 2023.
>
> [2] Edward J. Hu, Yelong Shen, Phillip Wallis, Zeyuan Allen-Zhu, Yuanzhi Li,
> Shean Wang, Lu Wang, and Weizhu Chen. Lora: Low-rank adaptation of
> large language models, 2021.

---

> ### Author Response · Authors · 2026-05-12
> **Reviewer 5wqL Requested Change 2**
>
> Requested Change: Let the LLM do prompt rewriting to have a detailed
> description of the layout and treat that description as the T2I prompt.
>
> Answer: For prompt rewriting, we utilize Prompt Enhancer [6], a strong,
> recent approach (CVPR 2026) that expands under-constrained prompts into
> varied text descriptions. PromptEnhancer does not natively support layout
> generation, so we feed the extended prompts into our LLM baselines and report
> the results.
>
> Model  |  CMMD ($\downarrow$)  |  FID ($\downarrow$)  |  KID ($10^{-2}$) ($\downarrow$)  |  HSPv2 ($\uparrow$)  |  Image Reward ($\uparrow$)  |  VQA ($\uparrow$) |
> | :--- | ---: | ---: | ---: | ---: | ---: | ---: |
> LLMBlueprint  | 0.21  | 1.18  | 0.71  | 0.25  | -0.29  | 0.90 |
> GPT4o with COT  | 0.17  | 1.29  | 0.87  | 0.25  | -0.45  | 0.89 |
> LayoutGPT  | 0.22  | 0.98  | 0.74  | 0.25  | -0.16  | 0.91 |
> LLMGroundedDiffusion  | 0.19  | 1.08  | 0.76  | 0.25  | -0.28  | 0.92 |
> Ranni  | 0.20  | 1.14  | 0.56  | 0.25  | -0.16  | 0.90 |
> Ours  | 0.03  | 0.17  | 0.16  | 0.25  | -0.32  | 0.88 |
>
> Our method outperforms all baselines on CMMD, FID, and KID, indicating better alignment with the training distribution. Although expanding the
> prompt generally improves baseline metrics over their default implementations,
> the main advantage of our method—accurately learning the full target distribution—remains.
>
> [6] Linqing Wang, Ximing Xing, Yiji Cheng, Zhiyuan Zhao, Donghao Li,
> Tiankai Hang, Jiale Tao, Qixun Wang, Ruihuang Li, Comi Chen, Xin Li,
> Mingrui Wu, Xinchi Deng, Shuyang Gu, Chunyu Wang, and Qinglin Lu.
> Promptenhancer: A simple approach to enhance text-to-image models via
> chain-of-thought prompt rewriting, 2025

---

> ### Author Response · Authors · 2026-05-12
> **Reviewer 5wqL Requested Change 3**
>
> Requested Change: Table 1 should include the threshold for what constitutes an effective difference for each metric, particularly as most HPS scores are
> around 0.25.
>
> Answer: We define effective-difference thresholds for generated image metrics by computing the mean absolute difference between metric evaluations on n = 10 sample sets drawn from the same distribution. Differences in the metric
> within this range are not considered meaningfully different.
> The metrics are computed we follow the sampling and evaluation procedure described in Section 4, paragraph: Generated Image Metrics. We sample
> ADE20K ground-truth images from 20 categories, (30 samples per category) to
> form the generated distribution. For the CMMD, FID, and KID metrics, we use
> images generated from BoxDiff, conditioned on ground-truth ADE20K layouts
> as the reference distribution, using the same categories and sampling scheme.
>
> Thresholds (written as plus/minus range):
>
> CMMD | FID | KID ($10^{-2}$) | HSPv2 | Image Reward | VQA |
> | ---: | ---: | ---: | ---: | ---: | ---: |
> ±0.01 | ±0.12 | ±0.07 | ±0.0003 | ±0.01 | ±0.002 |
>
> These thresholds do not change the ranking for CMMD, FID, nor KID, where
> our method remains best. HSPv2 slightly favors fine-tuned Ranni (0.259±0.0003
> vs. 0.248 ± 0.0003). See the revised Table 1 in the manuscript for the full results.

---

### Author Response · Authors · 2026-05-12
**Rebuttal Overview**

We thank the reviewers for their valuable feedback, and are happy that our work was found to:  address *a challenging research task with practical applications* (Reviewer 5wqL), *provide comprehensive empirical results, human evaluations, and visualization cases* (Reviewer 5wqL), be *relevant to researchers working on controllable generation, interpretable image synthesis, layout-conditioned diffusion models, and structured scene representation* (Reviewer mYfs), and provides *Extensive qualitative comparisons across different models (layout-to-image models) and generation stages* (Reveiwer CU4f).

We address the requested changes in comments below, and reupload a manuscript
with the requested changes marked in red. In particular, note that the new Table
1 contains the full updated metrics from experiments requested by reviewers.